# The Effects of Occupation, Education and Dwelling Place on Attitudes towards Animal Welfare in China

**DOI:** 10.3390/ani14050713

**Published:** 2024-02-24

**Authors:** Francesca Carnovale, Jin Xiao, Binlin Shi, David R. Arney, Clive J. C. Phillips

**Affiliations:** 1Chair of Animal Nutrition, Institute of Veterinary Medicine and Animal Sciences, Estonian University of Life Sciences, Kreutzwaldi 1, 51014 Tartu, Estonia; francesca.carnovale@emu.ee (F.C.); david.arney@emu.ee (D.R.A.); clive.phillips@curtin.edu.au (C.J.C.P.); 2College of Animal Science, Inner Mongolia Agricultural University, Zhaowuda Road 306, Hohhot 010018, China; shibinlin@yeah.net; 3Curtin University Sustainability Policy (CUSP) Institute, Curtin University, Perth, WA 6845, Australia

**Keywords:** animal welfare, attitudes, China, education, living place, occupation

## Abstract

**Simple Summary:**

Understanding people’s attitudes towards animals and their welfare is vitally important to enable governments to set a legal framework that meets people’s expectation for the way in which animals are cared for in their community. Little is known about attitudes towards animal welfare in China, the biggest producer of farm animals in the world. We surveyed 1301 people from different Chinese provinces to gauge their attitudes towards animal welfare. We found that high school leavers were more concerned about animal welfare than those who had obtained a university degree. Scientists were less supportive, and artists more supportive, of good animal welfare. Rural dwellers were less concerned about animal welfare than urban residents, with village residents in the middle.

**Abstract:**

Attitudes to animal welfare are not understood well in China, the country with the highest output of farm animals in the world. We surveyed attitudes of the public around China using a team of researchers to conduct individual interviews, with 1301 respondents in total. Contrary to results obtained in several other countries, high school leavers were more concerned about animal welfare than those who had obtained a university degree. We speculate that this may reflect the labour market currently existing in China, with limited opportunities for graduates. Scientists were less supportive, and artists more supportive, of good animal welfare. Urban dwellers were more concerned about animal welfare than rural residents, with village residents in the middle, which confirmed our theory that such a difference prevails in developing countries, where a large proportion of the rural population are involved in agriculture. It is concluded that education level, occupation and living place all have pronounced influences on attitudes to animal welfare in China, some of which follow international trends.

## 1. Introduction

A good understanding of the factors influencing attitudes towards animal welfare in key animal-producing countries and trading partners will help to establish animal management criteria that match people’s expectations. It will also assist in predicting the market for animal products and help to ensure that international trade in animal products is sensitive to the needs of the populations in exporting and importing countries. Many factors potentially influence people’s attitudes towards the welfare of animals, including their gender, age, level of education, especially any education related to animal welfare, and where people live. However, there have been few studies of public opinions, most surveys being of relatively small numbers of students, particularly veterinary students. Small numbers of respondents make it difficult to identify demographic effects accurately, especially if the surveying technique does not provide a representative sample. 

Level of education is one of the demographics for which effects on animal welfare attitudes are not well understood. In a study of food choices by 843 carnivorous Mexican respondents, those who had finished their education at elementary or secondary school were less supportive of choosing food that had been produced in a way that respected animals’ rights than students who had received higher education [1]. Similarly, Vanhonacker et al. [2] found greater concern in Flemish citizens about stocking density, pen size and group size in Flemish animal production systems if they were educated beyond 18 years. Social work students in a Hispanic-serving university near the US-Mexico border were more likely to purchase products labelled ‘not tested on animals’ if they were graduates [3], which could be related to their better purchasing power. Other surveys have focused on actions, not attitudes; a survey of 206 Hispanic respondents in the USA found that respondents who had been educated to elementary level were much more likely to chain their dogs, compared with those that had completed some high school education [4]. In another study, guardians of dogs in Brazil were more likely to have diseased dogs if welfare was not adequately considered and if they had either no formal education or were university graduates, compared with those educated to primary or secondary level [5].

Another of the major influences on people’s attitudes to animal welfare is where they live, but there is a diversity of evidence from published studies. In Mexico, living in a rural area is associated with less concern about animal welfare and greater acceptance of modern farming methods [6]. However, an earlier UK study of willingness to pay for high-welfare products found no difference between urban and rural respondents [7]. Similarly, Vanhonacker et al. [2] found no difference between rural and urban dwellers in level of concern about stocking density, pen size and group size in Flemish animal production systems. Boogard et al. [8], in a survey of 1074 Dutch respondents, found that respondents in highly urbanised areas (>2500 addresses/km^2^) believed that there was less difference between the value of human and animal lives, compared with respondents in urbanised areas (1500–2500 addresses/km^2^), and both these groups believed more than respondents in rural areas (<500 addresses/km^2^) that farm animals had a poor quality of life. The diversity of evidence on whether living place affects attitudes to farm animal welfare may relate to the degree of contact people have with farm animals and farmers [9]. In Australia, people living in large rural properties were more knowledgeable about chicken welfare but less sympathetic towards the welfare of the chickens than urban dwellers [10]. More contact could lead to sympathy for farmers, who may have to sacrifice the welfare of their animals to reduce the purchase price for consumers, e.g., in intensive animal husbandry practices. It could also lead to cognitive dissonance, people ignoring a problem that they are familiar with because they cannot see a solution. It might be concluded that in highly developed countries, such as the UK and Belgium, where rural dwellers are not necessarily in contact with farm animals any more than those in cities, there is little or no difference between people living in cities and rural areas, but in developing countries such as Mexico, there is likely to be significantly less concern about animal welfare amongst rural dwellers than those in the cities. These generalisations may extend to China, because urban stakeholders in the pig and poultry industries of this developing country are more likely to believe that they can make changes to animal welfare than those in rural districts [11]. They are also more likely to support the need for chickens to be happy, to be able to flap their wings and make nests.

A related factor influencing people’s attitudes to animal welfare is their occupation, especially if it involves animals. This has been little explored in surveys, in part because many are conducted with students. In one survey of attitudes of stakeholders in the livestock industries in SE and E Asia towards animal welfare, government officials were more likely to respect good animal welfare than those working directly with animals. Veterinarians are good at recognising poor welfare but less likely to think that people want it improved, compared with those working directly with the animals [11,12].

Attitudes towards animal welfare have been largely studied in Western countries. Of 2114 publications listed on the Web of Science database that contain the terms *animal welfare* and *attitude* in their topic, the geographic distribution is as follows: England 323, USA 323, Australia, 308, Germany, 194, Canada, 188, Italy, 138, Netherlands, 129, Scotland 108, Spain 97 and Brazil 90. There have been few studies in PR China, even though it is the world’s largest farm animal producer, with rapid changes underway in the intensification of production methods. Until recently, there was a common view that animal welfare was an unknown concept, or not well supported, in China relative to other countries [13]. Lately, however, a survey has suggested that empathetic attitudes towards animals in the Chinese public are widespread [14]. However, this online survey (which was conducted in July and in August 2021) was distributed via the social media app WeChat, and it is possible that those interested in animal welfare were more likely to have completed this survey. Thus, the results are not likely to be representative of public opinion [15]. A more representative survey was conducted by You et al. [16] in most Chinese provinces, but the survey was still dominated by better educated and young people, compared with the Chinese populace as a whole. Although the dominant view was an instrumental one, but with a strong respect for the ability of animals to live a natural life, the authors noted that attitudes varied among the different provinces. More educated respondents were more supportive of animal welfare. 

The Chinese aquaculture industry is the largest of any country in the world, but a survey of attitudes towards animal welfare in people connected with the aquaculture industry in China found that they mainly associated animal welfare issues with production of land animals, not aquatic animals [17].

In view of the paucity of information on the effects of these demographic factors in attitudes to animal welfare in China, we surveyed the public there, using students to help administer the survey. This paper focuses on the effects of education level, occupation and living place on respondents’ attitudes, the overall results [18] and effects of gender and age [19] having been previously reported.

## 2. Materials and Methods

The survey method was approved by the University of Queensland’s Human Research Ethics Committee (#2019001811) and has previously been reported [18] (Appendix A). Between August 2019 and September 2019, 217 undergraduate students from the Inner Mongolia Agricultural University delivered 2170 questionnaires in 23 directly governed provinces of the People’s Republic of China (hereafter China). Individual approaches were made to potential respondents in public places (e.g., shopping centres, streets, parks, squares, and marketplaces) as well as knocking door-to-door at people’s residences. A total of 1301 of these were returned and processed.

The format and content of the survey were initially prepared in English, then translated into written Chinese by a staff member from the Inner Mongolia Agricultural University who was bilingual in Chinese and English, and familiar with animal welfare terminology. During the presentation of the format verbal explanations were given if this was necessary. The Chinese version was then translated back into English and compared to the original version to confirm that the original meanings were maintained. The first section of the questionnaire (see Appendix A) concerned demographics: age, gender, level of education, professional background, religious affiliation, and place of dwelling. Regarding place of dwelling, the respondents were asked if, at the moment of the interview, they were currently living in a rural area, village, urban area or other. The classification of individuals as village residents, rural dwellers, or urban dwellers depended on factors such as the nature of their living environment, economic activities, and administrative designations. In this study, a resident in a village was identified as an individual where the primary place of living lay within the boundaries of a village. A rural dweller referred to someone living in a rural area characterized by open spaces, low population density, and an economy typically centred around agriculture or natural resources. An urban dweller resided in an urban area, such as a city or town, marked by higher population density, developed infrastructure, and diverse economic activities. This section also asked about the participant’s background and expertise in animal welfare, (e.g., Where did you learn about caring for animals? Who do you think is most responsible for the adequate care of animals?). Section 2 of the questionnaire was divided into four sets of questions with Likert-scale responses, asking how important it is that different types of animals are cared for (mammals, reptiles, birds, insects, pet animals, experimental animals, agricultural animals, stray animals and wildlife). Finally, they were asked why they thought animals should be cared for and which aspects of animal welfare were most important to them. Each response scale was transformed to a numerical scale from 1 to 5, in order from negative to positive affirmation, such as 1 (not sure) to 4 (many times); 1 (not at all) to 5 (to a great extent); 1 (not at all) to 5 (extremely); 1 (definitely not) to 5 (definitely);1 (5%) to 6 (>100%); 1 (very poor) to 5 (very good); 1 (much worse) to 5 (much better); 1 (strongly disagree) to 5 (strongly agree) and finally 1 (not at all important) to 5 (very important).

Survey respondents were asked to declare the highest educational level achieved, from the following options (with number of respondents): elementary school or below (131), technical college (146), middle school (160), high school (507), university undergraduate, hereafter ‘graduates’ (270) or university postgraduate, hereafter ‘postgraduates’ (86). They were asked if they were currently employed and, if they were, in which field, from the following options (with number of respondents): administration, agriculture, arts, construction, education, finance, government, health, mining, military, retail/sales, science, technology or other. The respondents who indicated they were in the military were subsequently removed. 

### Statistical Analysis

All analyses were conducted using the statistical package Minitab 18.0 (Minitab Version 18; Minitab Inc., State College, PA, USA). Descriptive statistics were generated and have been previously reported [18]. The effect of non-demographic questions in the survey were analysed by Ordinal Logistic Regression for ordered categorical dependent variables, in order to predict interactions between all demographic’ categories and groups of the category (gender, age, high education level, occupation and living place). Binary dependent variables were analysed by Binary Logistic Regression. This study reports only on the effects of education level, dwelling place and work field on respondents’ attitudes towards and knowledge of animal welfare. The group with the most responses was chosen as the reference group. Data are displayed as the means and differences in the number of responses between groups.

## 3. Results

A total of 1301 of the 2170 potential respondents completed the questionnaire, representing a 60% response rate. The distributions of respondents for the three demographic factors investigated are depicted graphically. Regarding highest education level achieved, the greatest number of respondents was received for high school leavers and the least for postgraduate (Figure 1). There were three non-respondents. Regarding dwelling place, over half of respondents were from urban areas, with least in rural areas, and just a few respondents declaring “Other” (Figure 2). There were two non-respondents. Regarding occupation, the greatest number of respondents classified as “Other” occupation, but almost as many were in agriculture (Figure 3). Mining and science had relatively few respondents. There were 11 in the military that were discarded and 35 non-respondents. 

### 3.1. Education Level

When asked if they had heard of animal welfare, high school leavers had heard of it more than those that had finished their education either earlier or later (Table 1). They were also more likely to say that they lived in harmony with animals, that caring for animals was important to them and that animal care should be taught in schools. University graduates were least likely to hold these pro-animal welfare attitudes. High and elementary school leavers and those with postgraduate qualifications were more willing to pay extra for products from animals that are better cared for, with postgraduates being willing to pay the most and undergraduates the least. Postgraduates were also more likely than high school leavers to say that the standard of animal care in China was worse compared to other countries. When asked about the importance of caring for different animal groups, high school leavers responded more positively than other education levels, with undergraduates responding the least positively for all animal groups except pet animals and wildlife. There was no significant difference in the responses for wildlife. High school leavers believed more strongly than undergraduates that people take care of animals for reasons of food safety, for the sake of the environment, because it makes them feel good, and for product quality or taste. They believed less strongly than undergraduates that human health was why people take care of animals. Similarly, high school leavers believed that all the requirements for animal care were more important than did undergraduates: these being adequate nutrition and drinking water, a comfortable environment, sufficient space, physical fitness and an opportunity to perform natural behaviours. High school leavers agreed more strongly than undergraduates, technical college and middle school leavers that farms with animals should be certified by animal protection organisations and that transportation time of live animals should be minimised, and more strongly than undergraduates that animals on farms should be provided with enjoyable experiences. High school leavers agreed more strongly than undergraduates, technical college and middle school leavers that farms with animals should be certified by animal protection organisations and that transportation time of live animals should be minimised, and more strongly than undergraduates that animals on farms should be provided with enjoyable experiences, that legislation should ensure that animal care is adequate and that animal protection organisations are important. Postgraduates believed more than high school leavers that it was acceptable to buy products of animals that have suffered if either the product quality was good enough or the price was low enough, that animals should be unconscious before they are killed and that animals should be killed before being cooked.

### 3.2. Occupation

Compared with the occupation “Other”, respondents who classified themselves as scientists were less likely to say that they had learnt about caring for animals from family and friends and more likely to believe that the standard of animal care in China was better than other countries (Table 2). Artists and to a lesser extent health workers, and to an even lesser extent agriculture, education and administration workers said they were willing to pay more for products from animals that were very well cared for compared with “Other” workers. Health workers were more likely to believe that it was important for animals to have control over their environment than “Other” workers.

### 3.3. Dwelling Place

Rural respondents were least likely to have heard of animal welfare, then village respondents, with urban respondents being most likely to have heard of it (Table 3). Urban respondents were most likely to say that they lived in harmony with animals, and that caring for animals was important to them as a person, then village respondents and then rural respondents. The urban respondents were more likely to have learnt about caring for animals from family and friends and animal protection organisations than those from rural or village dwelling places. Rural dwellers were least likely to have learnt from social media and most likely to have learnt from farmers. Urban dwellers were more likely to think that animal care should be taught in schools than those in villages and were more willing to pay for products from animals that were better cared for than those from villages or in rural areas. Urban dwellers thought it more important than rural dwellers that all animal groups, except mammals and reptiles, are cared for. No significant difference was recorded for mammals and reptiles.

## 4. Discussion

The distribution of respondents for nearly all categories in education level, occupation and dwelling place suggests that the sample was sufficient for the respondents to be reasonably representative of at least the provinces where most of the surveys were obtained. The province with most responses was Inner Mongolia. The Other category in Dwelling place was not commented upon as numbers for this were small. The response rate of 60% was less than the 70% recommended for surveys, but this is not a good indicator of non-response bias [20]. The attitudes described by our face-to-face survey are likely to be more representative than surveys that used social media to reach potential respondents. 

High school leavers (n = 570) consistently presented more pro-animal welfare views than people who had finished their education as undergraduates (n = 270). They were more aware of animal welfare and not only were their attitudes more supportive of animal welfare (caring for animal welfare was more important to them), their self-reported level of harmony with animals was greater than that of undergraduates. This suggests that they may care for animals better, or that they were more satisfied with the methods used by others to look after animals. This apparently conflicts with studies that have found greater support for food respecting animal rights in respondents who had received a higher education [1,3]. The latter could simply be a reflection of lower incomes of high school leavers, giving them less possibility to choose higher priced food that respected animal rights. Vanhonacker et al. [2] also found greater concern for animal welfare in survey respondents if they were educated past 18 years, which again could reflect greater purchasing power. However, in China, different circumstances may well prevail to make high school leavers support animal welfare more than those that have graduated from one of China’s many universities. First, civic moral education, which seeks to build the necessary knowledge and moral integrity to enable students to become responsible citizens, is much stronger in schools than at universities in China [21]. Even though moral education is now decentralized in China [22], and undergraduates would also have received the moral education in schools, their subsequent education in universities may have eroded their focus on morals instilled at school. Secondly, the income of workers that have received an undergraduate degree may not be higher than school leavers. The four or more years required for a Bachelor’s degree may disadvantage those in the job market in the early stages of their career. Also, now that the long period of economic growth in China has stagnated, the job market for graduates is limited and salary expectations are not usually met [23]. Although most studies have shown that better education equates with increased support for animal welfare (see Introduction), there is support for this contrary effect from studies with veterinary students. Veterinary students with a previous degree were less likely to choose the action choices which value life and bodily integrity of animals than students without previous degrees [24]. Verrinder and Phillips suggest that this was because the previous degrees were mainly in animal sciences, and students of this topic are known to be more accepting of harmful procedures to animals than those in other disciplines [25,26].

Respondents who had been postgraduates (n = 84) presented contrasting attitudes to both respondents with undergraduate degrees and high school leavers. They were more critical of China’s animal welfare standards than high school leavers, and thought animals should be killed properly, probably reflecting their greater knowledge. However, they were more prepared to buy products of animals that have suffered than school leavers, suggesting that even if the latter did not understand detailed aspects of animal welfare, they were more willing to support it. More ambitious people are likely to register for a postgraduate degree, which may make them less inclined to support initiatives that do not bring immediate benefit to them, such as improving animal welfare. 

We compared all occupation groups to the “Other” group, which, although it would have contained a variety of less popular occupations, presents a valuable comparison because it does not have the specific influences of individual occupations. Scientists were more likely to have learnt about animal welfare from other sources than family and friends, which is not surprising given the fact that their primary method of learning is through scientific endeavour. However, of greater concern is their defence of China’s standard of animal care, compared with other occupations. Scientists often use animals in their research and this may be essential for career progression, for which they tend to be highly motivated. Although they recognize questionable research practices in in a large proportion of other scientists, such as using research methods that harm animals, they themselves claim not to engage in these often [27]. The prevalence of questionable research practices depends on their discipline, with cognitive science and animal welfare science being two examples of disciplines where questionable research practices occur [27,28]. Caution is necessary because of the small number of scientists respondents. Artists, by contrast, were most willing to pay more for animals that were well cared for. Artists are inspired and attracted to animals, particularly in natural surroundings. Their strong sense of imagination enables them to capture the spirit, traits and personalities of animals, in a way that other people do not. The fact that health workers, to a lesser extent, were also prepared to pay more for well-cared-for animals and supported better control over their environment by animals is also not surprising. Doctors, nurses, pharmacists and paramedics are included in this group, which are known to have high levels of empathy [29]. Agriculture workers also were prepared to pay more, which probably reflects their understanding of the impact that poor living conditions has on animals in their industry, and their support for their own industry. 

The greater support for good animal welfare in urban respondents, more than those in villages and still more than those in rural dwellings, confirms previous research reviewed in the Introduction [2,6,7,8,9,10,11]. This concluded that rural dwellers in developing countries were more likely to be directly involved in, or at least closely connected with, agriculture compared with those in developed countries. As a result of cognitive dissonance and learning about animal welfare from their experience with production animals, rather than companion animals, they were less concerned with animal welfare than urban residents. China being a developing country, where a quarter of all workers are in agriculture, compared with, for example, the USA with just 2%, would be expected to have less concern for animal welfare in its rural population, compared with its urban population [30]. Rural dwellers were less likely to have heard of animal welfare, and, if they had, they were more likely to have heard about it from farmers rather than family, friends, animal protection organisations and social media. They were less concerned about caring for animals and less willing to pay for products from animals well cared for. The latter may be due to lower incomes and ability to pay. Village residents showed a level of concern somewhere between that of rural and urban dwellers, reflecting their limited contact with animals. 

## 5. Conclusions

In a survey of Chinese people’s attitudes towards animal welfare, we found that high school leavers were more concerned about animal welfare than those who had obtained a university degree. Scientists were less supportive, and artists more supportive of good animal welfare [27,28,29]. Rural dwellers were less concerned about animal welfare than urban residents, with village residents in the middle.

## 6. Limitations

The respondents in this study were more urbanised than the Chinese population overall. This could have been as a result of the sites chosen for the questioning. China is one of the largest countries in terms of population (1.4 billion) [31] and the relatively small number of respondents (1301) in our study may make it difficult to justify overarching demographic effects. The authors followed the literature and assessed the sampling error (and simple size representativeness) with two different equations [32] with a value of 0.085%, or an alternative equation [33] that provided a sampling error estimate of 2.77%. In both cases, the results showed that the sample size was adequate.

## Figures and Tables

**Figure 1 animals-14-00713-f001:**
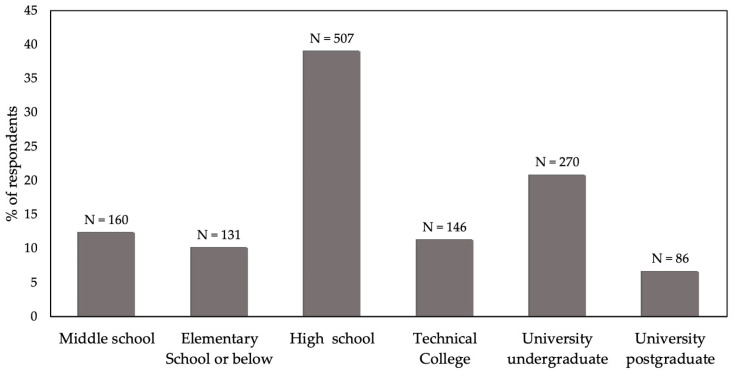
Highest education level in 1300 respondents in a survey of Chinese people’s attitudes to animal welfare.

**Figure 2 animals-14-00713-f002:**
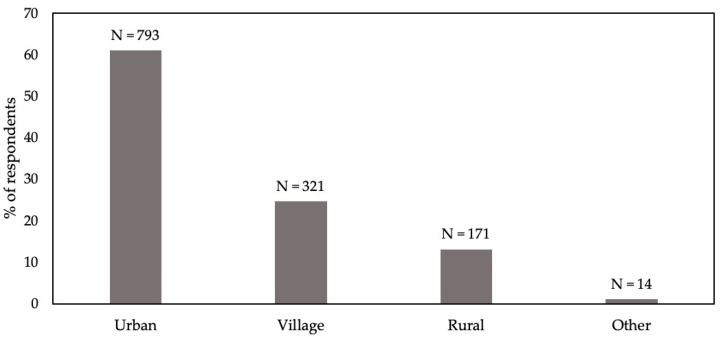
Dwelling place of 1299 respondents in a survey of Chinese people’s attitudes to animal welfare.

**Figure 3 animals-14-00713-f003:**
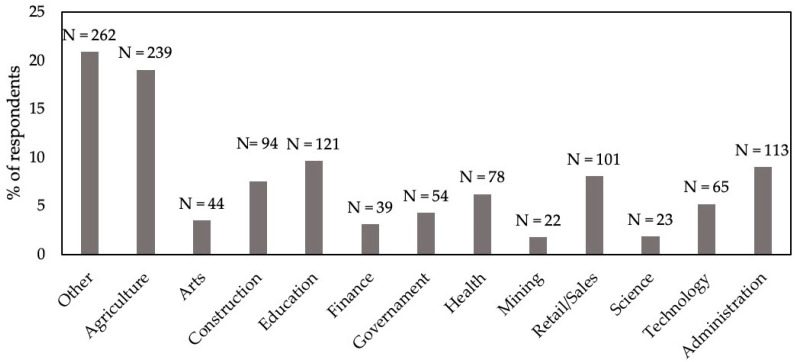
Occupation of 1255 respondents in a survey of Chinese people’s attitudes to animal welfare.

**Table 1 animals-14-00713-t001:** Level of education effects on respondents’ attitudes towards animal welfare in China. Mean responses on binary and 5-point ordinal scales are provided. Significant (*p* < 0.05) differences are presented between the reference education group High school (*n* = 507), and other groups: Technical college (*n* = 146), Middle school (*n* = 160), Undergraduate (*n* = 131), Elementary school or below (*n* = 270) and Postgraduate (*n* = 86).

Questions and Responses	High School Group Mean Score	Comparison Education Groups	Mean Scores	Odds Ratio	Coef.	95% CI		*p*-Value
						Lower	Upper	
Have you heard of the phrase ‘animal welfare’? 1 (Never) to 3 (Many times) ^1^	2.72	Technical college	2.37	2.05	0.71	1.38	3.04	<0.0001
		Middle school	2.19	3.82	1.34	2.53	5.77	<0.0001
		Undergraduate	2.10	4.35	1.46	2.69	7.03	<0.0001
		Elementary school	2.29	2.63	0.96	1.92	3.60	<0.0001
Do you live in harmony with animals? 1 (Not at all) to 5 (Extremely) ^2^	3.63	Technical college	3.06	1.89	0.63	1.31	2.74	0.001
		Middle school	2.96	2.04	0.71	1.40	2.97	<0.0001
		Undergraduate	2.90	2.15	0.76	1.39	3.32	0.001
		Elementary school	3.21	1.55	0.43	1.16	2.07	0.003
How important is caring for animals to you as a person? 1 (Not at all) to 5 (Extremely) ^2^	3.82	Technical college	3.17	2.60	0.95	1.79	3.78	<0.0001
		Middle school	3.21	2.29	0.82	1.56	3.35	<0.0001
		Undergraduate	2.99	2.90	1.06	1.87	4.51	<0.0001
		Elementary school	3.40	1.74	0.55	1.30	2.34	<0.0001
Do you think that animal care should be taught in schools? 1 (Definitely not) to 5 (Definitely) ^3^	3.43	Technical college	3.06	1.75	0.55	1.21	2.53	0.003
		Middle school	2.96	2.29	0.82	1.57	3.34	<0.0001
		Undergraduate	2.88	2.75	1.01	1.78	4.25	<0.0001
		Elementary school	3.07	1.79	0.58	1.34	2.40	<0.0001
Would you be willing to pay more for products from animals that are better cared for? (1 = yes, 2 = no)	1.28	Technical college	1.50					
		Middle school	1.49					
		Undergraduate	1.48					
		Elementary school	1.29					
		Postgraduate	1.28	1.13	0.13	1.07	1.22	<0.0001
How much more would you be willing to pay for a product from an animal very well cared for compared with the standard product? 1 (5%) to 6 (>100%) ^4^	2.41	Undergraduate	1.89	2.35	0.85	1.47	3.74	<0.0001
		Postgraduate	2.88	0.53	−0.62	0.34	0.83	0.005
How do you think the standard of animal care in China compares to other countries? 1 (Much worse) to 5 (Much Better) ^5^	2.38	Postgraduate	2.07	2.38	0.86	1.51	3.75	<0.0001
How important is it that the following animals are cared for? 1 (Not at all important) to 5 (Very important) ^6^ Mammals	4.27	Technical college	3.89	2.46	0.89	1.65	3.65	<0.0001
		Middle school	3.78	2.98	1.09	1.99	4.46	<0.0001
		Undergraduate	3.71	3.02	1.10	1.90	4.79	<0.0001
		Elementary school	3.95	1.97	0.67	1.44	2.69	<0.0001
Reptiles	4.17	Technical college	3.91	1.80	0.58	1.22	2.65	0.003
		Middle school	3.75	2.32	0.84	1.56	3.45	<0.0001
		Undergraduate	3.61	3.10	1.13	1.97	4.87	<0.0001
Birds	4.22	Technical college	3.95	2.09	0.73	1.42	3.09	<0.0001
		Middle school	3.86	2.28	0.82	1.54	3.38	<0.0001
		Undergraduate	3.77	2.80	1.03	1.78	4.41	<0.0001
		Elementary school	3.98	1.69	0.52	1.24	2.29	0.001
Insects	3.95	Undergraduate	3.57	1.86	0.61	1.20	2.87	0.005
Pet animals	4.35	Technical college	4.05	1.81	0.59	1.23	2.68	0.003
		Middle school	3.92	2.04	0.71	1.37	3.03	<0.0001
		Elementary school	4.07	1.56	0.44	1.15	2.12	0.004
Agricultural animals	4.36	Technical college	4.05	2.19	0.78	1.48	3.24	<0.0001
		Middle school	4.06	1.93	0.65	1.30	2.87	0.001
		Undergraduate	4.00	2.21	0.79	1.40	3.48	0.001
Stray animals	4.28	Technical college	4.04	1.74	0.55	1.18	2.55	0.005
		Middle school	3.94	1.79	0.58	1.21	2.65	0.004
		Undergraduate	3.83	2.32	0.84	1.48	3.64	<0.0001
Why do people take care of farm animals? Indicate how strongly you agree or disagree with the following reasons 1 (Strongly disagree) to 5 (Strongly agree) ^7^ It is important for food safety	4.19	Technical college	3.89	1.76	0.56	1.19	2.60	0.005
		Undergraduate	3.75	2.00	0.69	1.26	3.16	0.003
It is important for the sake of the environment	4.29	Undergraduate	3.83	2.31	0.83	1.46	3.65	<0.0001
It makes me feel good	4.1	Undergraduate	3.70	2.57	0.94	1.65	4.02	<0.0001
		Elementary school	3.87	1.55	0.43	1.14	2.09	0.004
		Postgraduate	4.37	0.46	−0.77	0.29	0.74	0.001
It is good for human health	4.13	Undergraduate	4.21	2.07	0.72	1.32	3.22	0.001
		Elementary school	3.91	1.55	0.43	1.15	2.09	0.004
		Postgraduate	4.36	0.50	−0.70	0.31	0.79	0.003
To improve product quality or taste	4.26	Undergraduate	3.93	2.11	0.74	1.34	3.31	0.001
		Elementary school	4.09	1.70	0.52	1.25	2.30	0.001
How important are the following conditions in animal care? 1 (Not at all important) to 5 (Very important) Species-relevant nutrition	4.31	Middle school	3.96	1.82	0.59	1.22	2.72	0.003
		Undergraduate	3.80	2.27	0.82	1.43	3.61	0.001
Access to drinking water	4.31	Undergraduate	3.82	2.77	1.01	1.75	4.39	<0.0001
A comfortable environment	4.36	Undergraduate	3.87	2.51	0.91	1.58	3.98	<0.0001
Space	4.38	Undergraduate	3.96	2.58	0.94	1.63	4.07	<0.0001
Physical fitness	4.43	Undergraduate	4.12	2.11	1.74	1.33	3.35	0.001
Opportunity to perform natural behaviours	4.30	Technical college	4.04	1.76	0.56	1.20	2.58	0.004
		Undergraduate	3.83	2.39	0.87	1.53	3.75	<0.0001
Indicate your level of agreement with the following statements 1 (Strongly disagree) to 5 (Strongly agree) ^7^ Farms with animals should be certified by animal protection organisations	4.15	Technical college	3.84	1.95	0.66	1.32	2.87	0.001
		Middle school	3.79	1.98	0.68	1.33	2.94	0.001
		Undergraduate	3.63	2.46	0.90	1.56	3.88	<0.0001
Transportation time of live animals should be minimised	4.17	Technical college	3.86	2.22	0.79	1.50	3.28	<0.0001
		Middle school	3.86	1.87	0.62	1.26	2.77	0.002
		Undergraduate	3.89	1.96	0.67	1.24	3.07	0.004
Animals on farms should be provided with enjoyable experiences	4.21	Undergraduate	3.98	2.08	0.73	1.32	3.27	0.002
It is OK to buy products of animals that have suffered if the product quality is good enough	3.11	Postgraduate	3.39	0.53	−0.64	0.34	0.81	0.004
It is OK to buy products of animals that have suffered if the price is low enough	3.00	Postgraduate	3.44	0.44	−0.81	0.29	0.69	<0.0001
Animals should be unconscious (stunned) before they are killed	3.96	Middle school	3.69	2.00	0.69	1.36	2.94	<0.0001
		Undergraduate	3.7	2.28	0.82	1.47	3.55	<0.0001
		Elementary school	3.78	1.59	0.46	1.18	2.14	0.002
		Postgraduate	4.16	0.51	−0.66	0.32	0.81	0.005
Animals should be killed before being cooked	4.11	Postgraduate	4.38	0.37	−0.98	0.23	0.60	<0.0001
It is important to have legislation that ensures animal care is adequate	4.37	Middle school	4.05	1.90	0.64	1.28	2.82	0.002
		Undergraduate	4.04	2.19	0.78	1.39	3.45	0.001
Animal protection organisations are important in ensuring animals are adequately cared for	4.39	Undergraduate	3.93	2.86	1.04	1.81	4.50	<0.0001

^1^: 1 = Never, 2 = A few times, 3 = Many times. ^2^: 1 = Not at all, 2 = Slightly, 3 = Moderately, 4 = Very much, 5 = To a great extent/Extremely. ^3^: 1 = Definitely not, 2 = Probably not, 3 = Possibly, 4 = Probably, and 5 = Definitely. ^4^: 1 = 5%, 2 = 10%, 3 = 20%, 4 = 50%, 5 = 100%, 6 = >100%. ^5^: 1 = Much worse, 2 = Somewhat worse, 3 = About the same, 4 = Better, and 5 = Much Better. ^6^: 1 = Not at all important, 2 = Slightly important, 3 = Neither important nor unimportant, 4 = Somewhat important, and 5 = Very important. ^7^: 1 = Strongly disagree, 2 = Disagree, 3 = Neither agree nor disagree, 4 = Agree, and 5 = Strongly agree. Coef. = coefficient, CI = Confidence Interval.

**Table 2 animals-14-00713-t002:** Effects of occupation on respondents’ attitudes towards animal welfare in China. Mean responses on binary, 5-point ordinal scales are provided. Differences are presented between the reference workgroups Other (*n* = 251), compared with other groups, Agriculture (*n* = 239), Arts (*n* = 44), Construction (*n* = 94), Education (n = 121), Finance (*n* = 39), Government (*n* = 54), Health (*n* = 78), Mining (*n* = 22), Retail/Sales (*n* = 101), Science (*n* = 23), Technology (*n* = 65), and Administration (*n* = 113).

Questions and Responses	Other Group Mean Score	Comparison Occupation Groups	Mean Scores	Odds Ratio	Coef.	95% CI		*p*-Value
						Lower	Upper	
Where did you learn about caring for animals? Family and friends (1 = no, 2 = yes)	1.37	Agriculture	1.35	0.79	−0.04	0.67	0.98	0.005
		Arts	1.31					
		Construction	1.44					
		Education	1.42					
		Finance	1.23					
		Government	1.35					
		Health	1.35					
		Mining	1.22					
		Retail/Sales	1.25					
		Science	1.13					
		Technology	1.27					
		Administration	1.36					
How much more would you be willing to pay for a product from an animal very well cared for compared with the standard product? 1 (5%) to 6 (>100%) ^1^	2.00	Agriculture	2.42	0.47	−0.74	0.32	0.69	<0.0001
		Arts	2.80	0.24	−1.43	0.12	0.46	<0.0001
		Education	2.46	0.39	−0.92	0.25	0.63	<0.0001
		Health	2.64	0.39	−0.93	0.23	0.67	0.001
		Administration	2.47	0.41	−0.89	0.25	0.66	<0.0001
How do you think the standard of animal care in China compares to other countries? 1 (Much worse) to 5 (Much Better) ^2^	2.39	Science	3.08	0.17	−1.74	0.08	0.40	<0.0001
How important are the following conditions in animal care? Indicate how strongly you agree or disagree with the following reasons 1 (Not at all important) to 5 (Very important) ^3^ Control over their environment	4.16	Health	4.46	0.44	−0.81	0.25	0.78	0.005

^1^: 1 = 5%, 2 = 10%, 3 = 20%, 4 = 50%, 5 = 100%, 6 = >100%. ^2^: 1 = Much worse, 2 = Somewhat worse, 3 = About the same, 4 = Better, and 5 = Much Better. ^3^: 1 = Not at all important, 2 = Slightly important, 3 = Neither important nor unimportant, 4 = Somewhat important, and 5 = Very important. Coef. = coefficient, CI = Confidence Interval.

**Table 3 animals-14-00713-t003:** Dwelling place effects on respondents’ attitudes towards animal welfare in China. Mean responses on binary, 5-point ordinal scales are provided. Differences are presented between the reference dwelling group Urban (*n* = 793), compared with other groups, Village (*n* = 321), and Rural (*n* = 171). Respondents who indicated Other (*n* =14) are not commented upon because of their small number.

Questions and Responses	Urban Group Mean Score	Comparison Dwell Groups	Mean Scores	Odds Ratio	Coef.	95% CI		*p*-Value
						Lower	Upper	
Have you heard of the phrase ‘animal welfare’? 1 (Never) to 3 (Many times) ^1^	2.62	Village	2.28	2.03	0.70	1.54	2.68	<0.0001
		Rural	2.04	3.96	1.37	2.67	5.86	<0.0001
		Other	3.14	0.20	−1.60	0.07	0.58	0.003
Do you live in harmony with animals?	3.52	Village	3.02	1.95	0.66	1.95	2.52	<0.0001
1 (Not at all) to 5 (Extremely) ^2^		Rural	2.97	1.89	0.63	1.89	2.68	<0.0001
How important is caring for animals to you as a person?	3.66	Village	3.28	1.79	0.58	1.39	2.32	<0.0001
1 (Not at all) to 5 (Extremely) ^2^		Rural	3.07	2.21	0.79	1.55	3.16	<0.0001
Where did you learn about caring for animals?								
Family and friends (1 = no, 2 = yes)	1.38	Village	1.29	0.79	−0.22	0.67	0.93	0.005
		Rural	1.30					
		Other	1.36					
Animal protection organisation (1 = no, 2 = yes)	1.98	Village	1.56	0.70	−0.35	0.54	0.90	0.004
		Rural	1.60					
		Other	1.43					
Social media (1 = no, 2 = yes)	1.30	Village	1.33	0.75	−0.27	0.63	0.90	0.002
		Rural	1.11					
		Other	1.29					
Farmer (1 = no, 2 = yes)	1.05	Village	1.03	1.69	0.52	1.28	2.22	<0.0001
		Rural	1.13					
		Other	1.35					
Do you think that animal care should be taught in schools?	3.33	Village	2.93	1.78	0.57	1.38	2.29	<0.0001
1 (Definitely not) to 5 (Definitely) ^3^								
Would you be willing to pay more for products from animals that are better cared for? (1 = yes, 2 = no)	1.35	Village	1.45	1.40	0.33	1.20	1.63	<0.0001
		Rural	1.61					
		Other	1.21					
How much more would you be willing to pay for a product from an animal very well cared for compared with the standard product?	2.34	Other	4.21	0.07	−2.63	0.03	0.19	<0.0001
1 (5%) to 6 (>100%) ^4^								
How do you think the standard of animal care in China compares to other countries?	2.57	Other	3.52	0.18	−1.71	0.06	0.51	0.001
1 (Much worse) to 5 (Much Better) ^5^								
How important is it that the following animals are cared for?	4.14	Rural	3.71	1.76	0.56	1.23	2.53	0.002
1 (Not at all important) to 5 (Very important) ^6^								
Birds								
Insects	3.93	Rural	3.46	1.98	0.68	1.39	2.81	<0.0001
Pet animals	4.25	Rural	3.77	0.64	0.86	1.64	3.41	<0.0001
Experimental animals	4.24	Rural	3.91	1.85	0.61	1.29	2.67	0.001
Agricultural animals	4.28	Rural	3.90	1.85	0.61	1.28	2.67	0.001
Stray animals	4.22	Rural	3.72	2.24	0.80	1.56	3.21	<0.0001
Wildlife	4.30	Rural	3.95	1.68	0.51	1.16	2.41	0.005
Why do people take care of farm animals? Indicate how strongly you agree or disagree with the following reasons	4.00	Other	4.57	1.19	0.17	0.83	1.70	0.005
1 (Strongly disagree) to								
5 (Strongly agree) ^7^								
It makes me feel good								
How important are the following conditions in animal care?	4.31	Rural	4.00	1.84	0.60	1.28	2.65	0.001
1 (Not at all important) to 5								
(Very important) ^6^								
Absence of pain								
Indicate your level of agreement with the following statements	3.33	Other	4.28	0.19	−1.6	0.07	0.55	0.002
1 (Strongly disagree) to								
5 (Strongly agree) ^7^								
Procedures performed on animals such as ear tags, castrations and tail breaks are acceptable for management								
It is OK to buy products of animals that have suffered if the price is low enough	3.06	Other	4.14	0.23	−1.47	0.08	0.64	0.005

^1^: 1 = Never, 2 = A few times, 3 = Many times. ^2^: 1 = Not at all, 2 = Slightly, 3 = Moderately, 4 = Very much, 5 = To a great extent/Extremely.^3^: 1 = Definitely not, 2 = Probably not, 3 = Possibly, 4 = Probably, and 5 = Definitely. ^4^: 1 = 5%, 2 = 10%, 3 = 20%, 4 = 50%, 5 = 100%, 6 = >100%. ^5^: 1 = Much worse, 2 = Somewhat worse, 3 = About the same, 4 = Better, and 5 = Much Better. ^6^: 1 = Not at all important, 2 = Slightly important, 3 = Neither important nor unimportant, 4 = Somewhat important, and 5 = Very important. ^7^: 1 = Strongly disagree, 2 = Disagree, 3 = Neither agree nor disagree, 4 = Agree, and 5 = Strongly agree. Coef. = coefficient, CI = Confidence Interval.

## Data Availability

The data presented in this study are available on request from the corresponding author.

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
