# Peer review of "The Effects of Occupation, Education and Dwelling Place on Attitudes towards Animal Welfare in China"

_animals, 2024, doi:10.3390/ani14050713_

Round 1
Reviewer 1 Report
Comments and Suggestions for Authors
Comments on the manuscript “The effects of occupation, education and dwelling place on attitudes towards animal welfare in China” submitted to the Animals
General comments
I appreciate the opportunity to review this exciting manuscript, which is a survey of the Chinese people's perception of animal welfare. The method was a direct and personal survey using 2,170 questionnaires applied by undergraduate students from the Mongolia Agricultural University. A total of 1,301 questionnaires were returned and processed. The authors concluded that school leavers were more concerned about animal welfare than those with a university degree. Scientists were less supportive, and artists were more supportive of good animal welfare. Rural dwellers were less concerned about animal welfare than urban residents, with village residents intermediate.
The manuscript is original and with a topic relevant to understanding the attitudes and thoughts on animal welfare in China. The text is easy to read and has good fluency. However, I think the manuscript has some weaknesses in methodology and theoretical basis.
On methodology
Considering that China is one of the largest countries in the world, in terms of the number of inhabitants and territorial area, a sample of 1,301 respondents is an underrepresentation. The authors understand that sample size is important, as stated in line 46: “Small numbers of respondents make it difficult to identify demographic effects accurately…”. However, this limitation of the study is not highlighted.
It is not described how the sample size was determined and how the questionnaires were distributed by region and sample group (rural village and urban dwellers), etc.
While the level of education and occupation are obvious, the place of residence is not defined. How to define a resident in a Village? What is a rural dweller? What is an urban dweller?
Apart from these general comments, I focus on some more detailed aspects of the text, which I present below.
Title
The spelling of the full stop is not included in the title.
Simple Summary, Abstract, and Keywords
It is good.
Introduction
Line 102: how and when was this research carried out on web of science?
Methods
Line 129: The research took place between August 2019 and August 2020. Since February 2020, there has been the COVID-19 pandemic, with many social restrictions, changes in attitudes, and the emergence of emotions such as anxiety among people in many countries. Nothing is mentioned about the visits, interviews, and movements of the interviewees during this period. I find it implausible that the COVID-19 pandemic did not influence the research.
Results
It is good.
Discussion
Line 287: I'm afraid I have to disagree that the respondents are reasonably representative of at least the provinces where most of the surveys were obtained. China is a huge and culturally diverse. There as many differences in culture and economy between the 23 Chinese provinces. The average of 56.56 questionnaires per province (1,301/23) is a number that may not reflect the distribution of the sample necessary to discuss the attitudes of the entire Chinese population.
Line 234: Could the willingness of Scientists to be less supportive of animal welfare be related to this period of the rush for vaccines and medicines to cure the sick? Can the authors comment on this atypical period in China?
Line 332-334: Which paper are the authors based on for their statement about "ambition"?
Line 349-354: Which paper are the authors based on for their statement about artist attitudes toward animals?
Conclusion
It is good.
References
It is good.
Author Response
Reviewer 1 General comments
Reviewer 1 Answer
I appreciate the opportunity to review this exciting manuscript, which is a survey of the Chinese people's perception of animal welfare. The method was a direct and personal survey using 2,170 questionnaires applied by undergraduate students from the Mongolia Agricultural University. A total of 1,301 questionnaires were returned and processed. The authors concluded that school leavers were more concerned about animal welfare than those with a university degree. Scientists were less supportive, and artists were more supportive of good animal welfare. Rural dwellers were less concerned about animal welfare than urban residents, with village residents intermediate.
The manuscript is original and with a topic relevant to understanding the attitudes and thoughts on animal welfare in China. The text is easy to read and has good fluency. However, I think the manuscript has some weaknesses in methodology and theoretical basis.
Thank you very much for your comment, and we appreciate your suggestions.
Considering that China is one of the largest countries in the world, in terms of the number of inhabitants and territorial area, a sample of 1,301 respondents is an underrepresentation. The authors understand that sample size is important, as stated in line 46: “Small numbers of respondents make it difficult to identify demographic effects accurately…”. However, this limitation of the study is not highlighted.
Sample size for Estimation.
Concerning the number of questionnaires; out of the 2,170 potential respondents, 1,301 completed the questionnaire, representing a 60.0% response rate. Our study indeed didn't involve a prior power analysis for selecting the sample size, but we acknowledge the importance of ensuring adequate statistical power in research design. According to the scientific literature, the practice of conducting post hoc power analysis has been subject to criticism due to the redundancy of p-values and effect sizes, which may lead to misinterpretation of results. For this reason, we decided not to implement a post hoc evaluation.
- https://www.graphpad.com/guides/prism/8/statistics/stat_why_it_isnt_helpful_to_compute.htm (
- https://pubmed.ncbi.nlm.nih.gov/11310512/
- https://www.acpjournals.org/doi/10.7326/0003-4819-121-3-199408010-00008?articleid=707593
- https://ladal.edu.au/pwr.html#Post-Hoc_Analyses
Nonetheless, the authors utilized an equation from the literature (Serdar et al., 2021) to assess the sampling error (and representativeness of our sample size), which yielded a value of 0.0858%. Additionally, an alternative equation (Lima et al., 2022) provided a sampling error estimate of 2.772436%.
- https://www.ncbi.nlm.nih.gov/pmc/articles/PMC7745163/
- https://periodicos.uninove.br/geas/article/view/20281/10038
Lastly, we acknowledge the limitations inherent in such approaches and recognize the importance of attention to sampling design and population representation. Thus, the authors modified Lines 355 of the manuscript to highlight this limitation and ensure the validity and generalizability of our findings.
Line 51: “Small numbers of respondents make it difficult to identify demographic effects accurately…
This comment is referred to other article concerning the number low of the categories such as number of veterinary students.
Lines 348 Added limitation section.
Lines 348 described how the sample size was determined.
The questionnaires were distributed as described in
Line 121 “in public places (e.g., shopping centres, streets, parks, squares, and marketplaces) as well as knocking door-to-door at people’s residences.” was not distributed around China in online format.
It is not described how the sample size was determined and how the questionnaires were distributed by region and sample group (rural village and urban dwellers), etc.
While the level of education and occupation are obvious, the place of residence is not defined. How to define a resident in a Village? What is a rural dweller? What is an urban dweller?
Line 128 authors added “The classification of individuals as village residents, rural dwellers, or urban dwellers depended on factors such as the nature of their living environment, economic activities, and administrative designations. In this study, a resident in a village was identified as an individual where the primary place of living lay within the boundaries of a village. A rural dweller referred to someone living in a rural area characterized by open spaces, low population density, and an economy typically centred around agriculture or natural resources. An urban dweller resided in an urban area, such as a city or town, marked by higher population density, developed infrastructure, and diverse economic activities.”
Title The spelling of the full stop is not included in the title.
The full stop is deleted, thank you very much.
Introduction
Line 102: how and when was this research carried out on web of science?
The authors added in Line 101 “However, this online survey was conducted in July and in August 2021 and was distributed via the social media app WeChat, and those interested in animal welfare were more likely to complete the survey.”
The reference for this is survey is given, line 103.
For Info: field surveys and online surveys conducted in July and August 2021. The selection of respondents followed the principle of random sampling. The online surveys were conducted through a professional web-based questionnaire platform. The questionnaire was distributed using a snowball sampling strategy through WeChat, a popular messaging app. The final proportion of paper and online questionnaires was approximately 3:1, with 2,795 paper questionnaires and 931 online questionnaires’.
Methods
Line 118: The research took place between August 2019 and September 2019.
Discussion
Line 287: I'm afraid I have to disagree that the respondents are reasonably representative of at least the provinces where most of the surveys were obtained. China is a huge and culturally diverse. There as many differences in culture and economy between the 23 Chinese provinces. The average of 56.56 questionnaires per province (1,301/23) is a number that may not reflect the distribution of the sample necessary to discuss the attitudes of the entire Chinese population.
Line 355: The authors added a limitation where is explained why is sustained that the sample size is representative.
Line 234: Could the willingness of Scientists to be less supportive of animal welfare be related to this period of the rush for vaccines and medicines to cure the sick? Can the authors comment on this atypical period in China?
Line 118 The authors mistaken for the date this survey was conducted before Covid. The research took place between August 2019 and September 2019.
Line 332-334: Which paper are the authors based on for their statement about "ambition"?
Authors added references (2, 6-11) line 330
Line 349-354: Which paper are the authors based on for their statement about artist attitudes toward animals?
Authors added references [27-29] line 346
Reviewer 2 Report
Comments and Suggestions for Authors
This is a valuable addition to the literature on attitudes toward animal welfare issues in a large but under-studied population. The review of the literature is comprehensive.
One language suggestion is to replace the term "high school leavers" with "high school graduates". The term "leaver" is likely to be unfamiliar to North American and European readers, although it seems to be is use in Australia. It is not a term I or my colleagues had encountered.
The data tables are informative but they can become unclear when the extend over more than one page. I suggest braking them into smaller tables or making sure that the column headings are repeated on each page so the reader does not have to flip back and forth.
Comments on the Quality of English Languagesee above note re: "leavers"
Author Response
Reviewer 2 Answer
Reviewer 2
This is a valuable addition to the literature on attitudes toward animal welfare issues in a large but under-studied population. The review of the literature is comprehensive.
One language suggestion is to replace the term "high school leavers" with "high school graduates". The term "leaver" is likely to be unfamiliar to North American and European readers, although it seems to be is use in Australia. It is not a term I or my colleagues had encountered.
Thank you very much for your comment, and we appreciate your suggestions.
However, we have thought about this and we still think that “leavers” is a less ambiguous term and should be understandable to English speakers. Even if “leavers” is not a common expression in North America (in Europe it is common), we think it is clear what is meant., If we change school “leavers” to school “graduates” this then becomes confusing as we also talk about undergraduates, postgraduates and graduates in regard to university level students.
The data tables are informative but they can become unclear when the extend over more than one page. I suggest braking them into smaller tables or making sure that the column headings are repeated on each page so the reader does not have to flip back and forth.
The authors understand the difficulties that readers can have, the authors will change the format of the tables before approving the publication of the article if, after the editing from the journal, the tables will still spread the tables over two pages.
Reviewer 3 Report
Comments and Suggestions for Authors
In my opinion, it would be worth specifying in the title of the article what group of animals was included in the welfare assessment. Were they all animals (including small pets) or only livestock animals (kept on farms). This suggestion results indirectly from the current formulation of the topic of the article.
In the final part of the Introduction chapter, the authors wrote what the focus of the research study was. However, I did not find a clearly formulated purpose of the research/study in this part of the article. I suggest writing the sentence "The purpose of the research/study was...". Moreover, when formulating the purpose/objectives of the research, it would be worth writing down what was the cognitive (scientific) goal and what was the utilitarian (useful) goal. Before stating the purpose of the work, it would be worth formulating the research problem. I think that based on the review of the state of knowledge presented in the Introduction of the article, a research problem can be easily formulated. I suggest that in the summary of the state of knowledge review, you simply write the sentence: "The research problem is...". Moreover, the research problem can be associated with presenting a gap in the current state of knowledge. This can also be done on the basis of the information presented in the Introduction about the current approach and assessment of animal welfare.
I would like to ask whether preliminary research was carried out before the main study using a survey? What I mean is to give the survey to a smaller group of people to see if there are any problems with understanding the questions and interpreting them clearly. Thanks to such a preliminary examination, it is possible to verify the formulation of the questions and the problems posed in them.
It would be worth writing about any possible problems encountered when respondents completed the surveys.
When collecting survey data, was it possible to additionally explain the essence of some questions? For example, the question "Do you live in harmony with animals?" in my opinion, it required clarification of what was meant by "harmony".
The calculation of the data labels in Figure 1 results in a total of 1,300. However, the figure's caption mentions the number of 1,298 respondents. These data probably need to be corrected, unless I missed something in the analysis of the information regarding Figure 1. In Figures 2 and 3, the sums agree with the data in the figure captions.
In my opinion, the Conclusions could be more developed. In this part or at the end of the Discussion chapter, the authors could write about the prospects for further research taking into account animal welfare in the opinion of respondents. In the discussion of the research results, it would be possible to expand on the issue of specialization of students who comment on animal welfare, which has an impact on the answers given. Such an approach, taking into account various specializations represented by students, can be found in the publication "Understanding animal welfare by students and graduates of different studies". The discussion may also point to the possibility of developing a research idea in the studied regions, consisting in assessing the perception of animal welfare as a result of visiting farms by urban citizens. Details of this approach to research can be found in the study "What difference does a visit make? Changes in animal welfare perceptions after interested citizens tour a dairy farm”. I think that showing - based on the examples provided - the perspectives for the development of research on welfare in China will be a valuable summary of the article.
Author Response
Reviewer 3 Answer
In my opinion, it would be worth specifying in the title of the article what group of animals was included in the welfare assessment. Were they all animals (including small pets) or only livestock animals (kept on farms). This suggestion results indirectly from the current formulation of the topic of the article.
Thank you very much for your comment, and we appreciate your suggestions.
The authors would like to keep this title, all animals were considered. Is clearer in the rest of the article.
In the final part of the Introduction chapter, the authors wrote what the focus of the research study was. However, I did not find a clearly formulated purpose of the research/study in this part of the article. I suggest writing the sentence "The purpose of the research/study was...". Moreover, when formulating the purpose/objectives of the research, it would be worth writing down what was the cognitive (scientific) goal and what was the utilitarian (useful) goal. Before stating the purpose of the work, it would be worth formulating the research problem. I think that based on the review of the state of knowledge presented in the Introduction of the article, a research problem can be easily formulated. I suggest that in the summary of the state of knowledge review, you simply write the sentence: "The research problem is...". Moreover, the research problem can be associated with presenting a gap in the current state of knowledge. This can also be done on the basis of the information presented in the Introduction about the current approach and assessment of animal welfare.
Thank you very much the authors, please are you sure that is not clear from Lines 111-114
I would like to ask whether preliminary research was carried out before the main study using a survey? What I mean is to give the survey to a smaller group of people to see if there are any problems with understanding the questions and interpreting them clearly. Thanks to such a preliminary examination, it is possible to verify the formulation of the questions and the problems posed in them.
It would be worth writing about any possible problems encountered when respondents completed the surveys.
When collecting survey data, was it possible to additionally explain the essence of some questions? For example, the question "Do you live in harmony with animals?" in my opinion, it required clarification of what was meant by "harmony".
Thank you for this, has was done and the questionnaire controlled from the ethics committee, but verbal explanation was given if it was necessary. The authors have added a sentence in Line 125
The calculation of the data labels in Figure 1 results in a total of 1,300. However, the figure's caption mentions the number of 1,298 respondents. These data probably need to be corrected, unless I missed something in the analysis of the information regarding Figure 1. In Figures 2 and 3, the sums agree with the data in the figure captions.
Thank you very much to notice this, the authors have replaced with the correct value.
In my opinion, the Conclusions could be more developed. In this part or at the end of the Discussion chapter, the authors could write about the prospects for further research taking into account animal welfare in the opinion of respondents. In the discussion of the research results, it would be possible to expand on the issue of specialization of students who comment on animal welfare, which has an impact on the answers given. Such an approach, taking into account various specializations represented by students, can be found in the publication "Understanding animal welfare by students and graduates of different studies". The discussion may also point to the possibility of developing a research idea in the studied regions, consisting in assessing the perception of animal welfare as a result of visiting farms by urban citizens. Details of this approach to research can be found in the study "What difference does a visit make? Changes in animal welfare perceptions after interested citizens tour a dairy farm”. I think that showing - based on the examples provided - the perspectives for the development of research on welfare in China will be a valuable summary of the article.
Yes, these are good points, but are somewhat beyond the limits of the questions asked in the survey and the classifications of the respondents. We did not collect separately the specialist subjects taken by the students so cannot comment on this with any justification. Likewise, visiting farms. And this too is fraught with difficulty, depending as it does, on the previous level of understanding of farming practices, preconceptions and willingness to learn. A good topic for study certainly, but not anything we can unpick from our research results.
Round 2
Reviewer 1 Report
Comments and Suggestions for Authors
I appreciate the authors' responses. The explanations regarding the doubts I raised previously are clear. The changes made in the text were important. After reading the new version of the manuscript, I am convinced there has been a substantial improvement, to the point that the manuscript is ready for publication.